# Health Care Needs in School-Age Refugee Children

**DOI:** 10.3390/ijerph16214255

**Published:** 2019-11-01

**Authors:** Anders Hjern, Stefan Kling

**Affiliations:** 1Centre for Health Equity Studies (CHESS), Karolinska Institutet/Stockholm University, 106 91 Stockholm, Sweden; 2Clinical Epidemiology, Department of Medicine, Karolinska Institutet/Stockholm University, 106 91 Stockholm, Sweden; 3Sachsska Children’s Hospital, 116 31 Stockholm, Sweden; 4Department of Child and Adolescent Psychiatry, Lund University Hospital, 221 85 Lund, Sweden; stefan.kling@skane.se

**Keywords:** health examination, health assessment, children, health care, convention on the rights of the child, migration, primary care, asylum seekers, refugee

## Abstract

Most European countries have systematic health assessments of refugees with a main focus on infectious diseases. The aim of this study was to describe the broader health care needs identified in newly settled refugee children in a school health setting. The study population consisted of all 609 recently settled Non-European refugee and asylum-seeking children in the age range 6–15 years who were enrolled in the schools of Malmö, Sweden during the autumn semester of 2015, of which 265 had arrived in Sweden unaccompanied. The data were collected in a structured routine intake interview by an experienced school nurse. Almost half of the children had obvious untreated caries. For the unaccompanied children, prominent mental health needs were present in almost one in three. Previously unidentified vision and/or hearing problems were identified in one in ten and around 5% had a daily medication, and 4.5% of the unaccompanied children and 1.2% of the accompanied children were judged to be in need of immediate care and were referred accordingly. Newly settled refugee children in northern Europe have considerable health care needs apart from communicable diseases. School health services have a unique platform to identify and initiate this care.

## 1. Introduction

During the years 2015–2017, 2.5 million asylum seekers were reported in the 28 EU member states alone, including almost one million children below 18 years of age, of whom 200,000 arrived unaccompanied by a parent or caregiver. Most European countries provide systematic health examinations of asylum seekers and sometimes also other categories of migrants [1]. In most eastern European countries and Germany, this health examination is mandatory with a focus on infectious disease; while in the rest of western and northern Europe it is mostly voluntary and also tries to identify individual health care needs. Epidemiological studies of asylum-seeking and newly settled refugee children in northern Europe have mostly focussed on infectious disease [2,3] and mental health problems [4], whereas there is a dearth of studies with a broader perspective of health care needs [5].

During the autumn of 2015, Sweden received around 150,000 asylum applicants, including more than 35,000 applications from children unaccompanied by a caretaker. In the city of Malmö, as in many other Swedish municipalities, unprecedented numbers of asylum-seeking children were accommodated in the school system that autumn. In the Malmö school system, the health care needs of recently arrived children with an origin outside Sweden are assessed by an experienced nurse. To facilitate the provision of adequate health care resources for these children, the city decided to analyse the routine data collected in this assessment. This article presents the results of that analysis of health care needs, including vaccinations.

## 2. Methods

During the autumn of 2015, 639 children in the age range 6–15 years entered the school system in Malmö with a Non-European nationality that made it probable that they were either asylum seekers or children in refugee families, according to the statistics produced by the Swedish Board of Migration. All children who entered a school in the city of Malmö from a foreign country were introduced into the school health services by a special introduction unit. The nurse, who served this unit, invited all newly settled foreign-born children and their guardians to a health assessment with the aim of identifying health care needs. For unaccompanied children, the invitation was sent to the guardian provided by the city of Malmö. In the letter of invitation it was made clear that their participation in the assessment was voluntary, but all invited children and their guardians chose to participate. Before the assessment was started, the children were informed that they were free to refuse all or any part of the assessment. 

The structured health assessment was made in the presence of the guardian and included an interview in Swedish through an interpreter, a superficial examination of teeth with a flashlight looking for open cavities and discoloured teeth, an examination of anthropometric measurements plotted on a height and weight chart, a screening for scoliosis with a scoliometer [6], and an examination of the skin of the abdomen, back and extremities. A structured sheet, that was also used to record the findings, guided the assessment (see Appendix A). Screening for infectious disorders was conducted at a separate county council unit outside of the school health system and was therefore not included in this school health assessment. 

Questions were posed regarding acute symptoms (diarrhoea, jaundice, cough, fever, skin problems, fatigue, pallor, nightly sweat) and longstanding health problems (stunted growth, disabilities and medications). Mental health problems were asked for through open questions about the presence of symptoms associated with traumatic events and sleeping disturbances were explicitly asked for and recorded. In the few cases where the children showed signs of uneasiness when sensitive questions were asked, that part of the interview was abbreviated, to be completed later when the child was more settled. To be recorded as yes, mental health problems, including sleeping disturbances, had to be severe enough that the nurse judged them to impair the well-being of the child on a daily basis. Eyesight was examined with a HVOT table if the child was not yet literate and otherwise with a Snellen table, with a vision tested as 0.8 or lower being considered a deficiency, as suggested by the Swedish national guidelines for the school health services [6]. Hearing problems were screened for with audiometry where a measurement of 25 dB on two or more frequencies was considered abnormal [6]. Vaccination history was looked for in documents, and if such documents were not available, structured questions were asked.

The health care needs identified were addressed firstly by the school health team, including psychoeducational advice and a follow-up by them for children with mental health problems. When more specialised services were needed, referrals were made according to the needs identified, including child and psychiatric services when mental health problems were found to be severe. The main findings from the health assessment were compared between unaccompanied and accompanied children in statistical chi-square analyses, conducted in SPSS version 25.0 (IBM, Armonk, NY, USA). In accordance with Swedish research legislation, this study, based on anonymised statistics from routine health care, was not passed through an ethical committee. 

## 3. Results

Table 1 presents the characteristics of the study population. Around 60% of the children had an origin either in Afghanistan or Syria, with the Afghans making up 61% of the unaccompanied children. The mean age was 10.3 years for the accompanied children and 13.4 years for the unaccompanied. As many as 87% of the unaccompanied children were boys, while gender was more evenly distributed among the accompanied children. 

As Table 2 indicates, 4.5% of the unaccompanied children and 1.2% of the accompanied children were judged to be in need of immediate care and were referred accordingly, while 38.2% of the unaccompanied children and 19.2% were judged to need to see the school health team in more depth. Dental health care needs were abundant with almost half of the children having untreated caries. For the unaccompanied children, mental health needs were prominent with post-traumatic stress and sleeping problems identified in 21.8% and 32.8% compared with 5.8% and 21.8% in the accompanied children. Impaired vision was identified in around 18% of the children, while impaired hearing was more common in the unaccompanied children, 12% vs. 6% in the accompanied children. 

Only 27% of the accompanied children and 1.5% of the unaccompanied children had documents that could verify their vaccination history, but as many as 75% of the accompanied children and 42% of the unaccompanied were judged in interviews to have had “all vaccinations” according to the vaccination schedule in the country of origin.

## 4. Discussion

The newly settled refugee children in the school system in Malmö were found to have significant needs for dental health care, mental health care, care for disabilities and chronic disorders. The needs identified were particularly prominent with regards to dental health and mental health.

The high rates of dental health care are consistent with higher risk findings in the previous studies of the dental health of migrant children in Sweden [7], although the rates found in this study are extremely high when the crude nature of the examination is considered. Previous Swedish studies have suggested that irregular tooth brushing and caries-promoting food habits contribute to the higher risk of caries in the children in migrant families compared to the children with Swedish-born parents [8]. For newly settled refugee children, it seems probable that the special food situation and the lack of dental health care in the war stricken countries of origin and during the journey to Sweden may have contributed to the high caries rate [9]. Because of the high caries prevalence in many of the migrant-dense schools in Malmö, interventions to prevent caries by administering fluoride salt to the students have been tried, but without much success to date [10].

The high prevalence of introverted mental health problems in the newly settled refugee children identified in this study is very much in line with previous studies [11]. Psychological trauma, often associated with war and persecution in the country of origin or the events of the journey is an important risk factor for the poor mental health and well-being of the newly arrived migrant children [11]. 

The higher rates of mental health problems in the unaccompanied children have also been shown in previous European studies [12,13], although the population of unaccompanied minors in this study was younger than in these studies. This underlines the greater need for psychological support that unaccompanied children have because they lack the support of their families. 

Interventions based on psychoeducational principles have been developed to help migrant children cope with their symptoms [14]. A growing body of evidence and experience has shown that schools play a critical role in protecting and promoting the health of migrant and refugee children [15]. Successful school-based mental health prevention requires professionals trained in cultural competence, who understand the mental health needs and risks of migrant children, and who are able to adapt the learning program to the needs of the individual child and family. School-based programs for the prevention of mental health problems in refugees have also been developed, and include programs that focus on trauma-associated symptoms, a program using particular treatment modalities to promote child mental health, and programs that focus on promoting a health adaptation to the host society in a holistic manner [16]. 

The lack of documentation about vaccinations demonstrated in this study, constitutes a major challenge for school health services in their ambition to provide vaccinations on par with the national guidelines in the receiving countries. Blood tests for vaccine antibodies are available but are usually too expensive for routine use in school health services. The vaccination rates found in the refugees this study are similar to those found in a recent study of recently settled children in Berlin [17] based on a similar type of information. Several studies of vaccine antibodies in the blood samples of refugee children in Europe during recent years have confirmed this comparatively low vaccination coverage for hepatitis B, measles, mumps, rubella, tetanus and diphtheria [18]. 

## 5. Strengths and Limitations

The origin of the children in this study was quite similar to the general origin of asylum seekers in the EU during 2015 according to Eurostat [19], with Syria and Afghanistan as the two dominant countries. It thus seems reasonable to generalise the more general patterns of health care needs in this study to the broader population of the newly settled Non-European refugee populations of school children in northern Europe during recent years, although specific rates might have varied considerably between populations over time and between sites. The routine data from school health care had some obvious limitations regarding the quality of the data, particularly with respect to mental and dental health, where the methods used were very crude. It seems more than likely that standardised instruments for mental health assessment would have identified more children with mental health problems, and more children with untreated caries would certainly have been identified in an examination by a dentist with the proper instruments. The lack of documentation of the vaccinations received prior to arriving in Sweden makes it necessary to interpret the findings here with great caution. The studies of vaccine antibodies in the representative samples of refugees should be considered to inform the vaccination strategies of health services.

## 6. Conclusions and Implications

According to Article 13 of the Reception Conditions Directive, EU Member States may require health assessments of applicants for international protection on public health grounds. There is, however, no obligation to undertake such assessments [20]. Article 24 of The United Nations Convention of the Rights of the Child states that all children, including all categories of migrant children, have the right to the highest attainable standard of health care. An assessment of health care needs can be an important part of the fulfilment of this obligation for migrant children [21]. This study showed that this health assessment needs to have a broad perspective, covering mental health, disabilities, dental health and chronic disorders apart from infectious disorders and vaccinations. School health services have unique potential to identify these broad health care needs within their routine services and to initiate the care needed [22].

## Figures and Tables

**Table 1 ijerph-16-04255-t001:** Characteristics of the study population of school-age refugees.

Nationality	All	Accompanied			Unaccompanied		
	***N***	***N***	Mean Age Years	Male %	***N***	Mean Age Years	Male %
Afghanistan	184	22	11.1	45	162	13.8	93
Ethiopia/Eritrea	14	1	14.0	100	13	14.6	77
Iraq	51	39	9.8	67	12	12.9	92
Pakistan	13	12	10.4	50	1	10.1	100
Palestine/Stateless	73	59	10.1	49	14	12.4	57
Somalia	38	18	10.3	72	20	14.0	60
Syria	185	145	10.1	53	40	11.9	87
Other	51	48	10.6	56	3	11.7	100
All	609	244	10.3	54	265	13.4	87

**Table 2 ijerph-16-04255-t002:** Health problems in the population of school-age refugees.

Problem	Symptom	All (*N* = 609)	Accompanied (*N* = 344)	Unaccompanied (*N* = 265)	
		%	%	%	*p*-Value ^1^
Health care needs					
	Immediate referral	2.6	1.2	4.5	< 0.01
	School health team	27.6	19.2	38.5	< 0.001
Mental health					
	Sleeping problems	23.0	15.4	32.8	< 0.001
	Symptoms associated with post traumatic stress	12.8	5.8	21.8	< 0.001
	Other mental health problems	4.1	2.6	6.0	< 0.01
Disabilities/ Chronic disorder					
	Daily medication	4.6	4.7	4.5	ns
	Impaired vision	18.6	17.5	18.8	ns
	Impaired hearing	8.2	5.5	11.8	< 0.05
	Other disabilities	1.8	2.0	1.5	ns
Dental health					
	Pain	4.1	4.1	4.2	
	All untreated caries	48.1	47.6	48.7	ns

^1^ = *p*-value of the difference between accompanied and unaccompanied in the chi-square test.

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
