# Peer review of "Health Care Needs in School-Age Refugee Children"

_ijerph, 2019, doi:10.3390/ijerph16214255_

Round 1
Reviewer 1 Report
Line 16... 6-15 (years)? .. should be specific
Line 17-' inscribed' should be changed to 'enrolled'
Line 22- author can consider changing 'had daily medication' to read ' required taking daily medication'
Line 38- given the context of the sentence 'death' should read 'dearth'
Line 53- Superficial assessment of teeth- a little more information on what was assessed should be included (levels of plaque, open cavities, discoloured teeth, broken restorations?)
Lines 60-64 on mental health assessment can be written e.g.... " Mental health problems were assessed using open questions where the presence of symptoms associated with traumatic........were recorded"
Line 64- what kind of 'table' did you use to assess eyesight?
In table 1 it is unclear what the 3rd column represents
Author Response
Thank you so very much for all these suggested helpful changes. We have changed the manuscript accordingly.
Reviewer 2 Report
Major revisions can be viewed in the attached document.

Author Response
INTRODUCTION
‘caretaker’ means different things and I suggest using one or even two terms (like guardian, parent or caregiver) to be more universal.
Comment. We have changed the text accordingly.
For the following sentence it may probably worthwhile to explain the differences between countries (by naming them and giving examples): “In most eastern European countries and Germany this health examination is mandatory with a focus on infectious disease; while in the rest of western and northern Europe it is mostly voluntary and also tries to identify individual health care needs.” Alternatively, rather speak in greater detail about the country in question as this sentence isn’t really adding much to explain the context.Comment: The referenced article describes the situation in 30 EU and EES countries. It would be beside the point to give a detailed description of all those countries here, anyone interested can read that article. Nonetheless, these screening policies provides the main narrative for this study, the narrative that we use to interpret our results. Thus, we think it would be great loss to take it out.
In line 38 I think the word ‘death’ should be ‘dearth’.Comment: We have changed accordingly.
Can the following sentenced be referenced and have some %s added? “The city of Malmö was the point of entry for most of these asylum seekers, and although most asylum seekers continued to other parts of Sweden, many stayed in Malmö.”Comment: That would indeed have been useful information, but there is no available statistics over point of entry to Sweden of asylum seekers at this time, since the borders were open. The paragraph has been rephrased as:
“During the autumn of 2015, Sweden received around 150 000 asylum applicants, including more than 35 000 applications from unaccompanied children. In the city of Malmö, as in many other Swedish municipalities, unprecedented numbers of asylum seeking children were accommodated in the school system that autumn. In the Malmö school system, the health care needs of recently arrived children with an origin outside Sweden are assessed by an experienced nurse.”
METHODS
An age range in the first sentence of the methods will be a helpful addition
Comment: This has been added.
The fact that the study was not passed by an ethical committee would be less concerning if there was a bit more detail. I strongly recommend that the following points be mentioned/addressed: Were children allowed to refuse the screening? Were children observed for signs of uneasiness during the interviews (for example, if children were obviously uncomfortable and cried or perspired)? If yes (to the above question), were children ever exempt from the screening? How was the body examination conducted? Were children tested by nurses of the same sex as them? A body examination can be regarded as invasive and there is very little detail about this. There is no mention of which language was used to conduct the interviews. How did the nurse decide which children’s’ mental health symptoms were severe enough? What table was used to examine eyesight? What audiometry test was used to examine hearing? There is very little mention about how the children’s rights as human beings were regarded. Did they ever provide consent? It is obvious that the health screening is in the best interests of the refugee children, but details of these sensitive tests should be provided.Comment: The methods section has been thoroughly revised in light of these valuable comments
RESULTS
In Table 1, why are Ethiopia and Eritrea combined? Why are Palestine and Stateless combined?
Comment: Until 1992, Eritrea was a part of Ethiopia. There are many shared cultural similarities between the two countries, including language and religion. The large majority of stateless refugees in Sweden have a Palestinian heritage.
In Table 1, there is an “N” missing above the numbers for accompanied children. Comment: Has been changed accordingly. A definition of ‘sleeping problems’ would be helpful in the methods.Comment: Judged to impair daily functioning was the definition of all mental health problems, including sleeping problems. This has been clarified in the revised methods section.
In Table 2, unaccompanied is spelled incorrectly. Comment: Has been changed accordingly.In Table 2, a column for ‘all’ would be helpful. Comment: Has been changed accordingly.
DISCUSSION
Measles is indeed a very big concern but the lack of vaccination documentation does not mean that the children have not been vaccinated. The discussion is very short and to focus exclusively on vaccines is doing the paper a disservice. The second paragraph should be deleted in its entirety and framed in such a way that does not assume that refugee children are responsible for the outbreaks of measles. This is how it reads at the moment.
Comment: This paragraph has been completely rewritten, deleting any wording that could give the impression that refugee children are responsible for measles outbreak
A worthwhile comment to make is that health screening in refugee children should focus on finding ways to screen children effectively for proof of vaccinations. Comment: this has been more clearly spelled out now. The results report a high proportion of children struggling with sleep and stress – this is going to directly impact the children’s health and well-being (as well as their ability to perform at school) and a lot more discussion is required on this topic.Comment: The discussion has been developed along these lines.
The results report a very high proportion of children with dental problems. It is therefore worthwhile to discuss how this can be improved (better nutrition, education about developing good dental hygiene habits etc.). It may also be worth investigating the cause of this.Comment: The discussion has been developed along these lines.
There are stark differences between accompanied and unaccompanied children. This is important and should be discussed in greater detail.Comment: The discussion has been developed along these lines.
STRENGTHS AND LIMITATIONS
An obvious limitation is that the majority of the children did not have vaccination documentation. This cannot be stressed enough. The lack of vaccination documentation does not mean that children have not received vaccinations. This needs to be mentioned.
Comment: see above.
It is absolutely not reasonable to generalise the findings of this study. There may be similarities but generalising findings is not correct.Comment: We tend to disagree here. We believe the general patterns are generalizable to to other refugee children in northern Europe at the time of the data collection. The history of refugees, however, tend to vary greatly over time and we agree completely that the results of this study are not generalizable to other time periods or to middle and low income settings. This has been clarified in the revised article.
IMPLICATIONS
Is it worth mentioning that perhaps health screening should be conducted by nurses but that where there is uncertainty, a specialist should be consulted? For example, in a situation where a child does not have vaccination documentation, a
2
doctor specialising in infectious diseases examines the child? Or for a child with many caries, a dentist is consulted for a more thorough inspection? Or a child with obvious mental health disorder symptoms is referred to a psychiatrist or psychologist? Comment: We believe such recommendations go beyond the results of this study.
Another implication of this study is that refugee children are at an increased risk in terms of their health and well-being. Greater attention should be paid to their needs as the environment from which they came as well as the stress of migration is obviously having an influence on their health, especially when they are unaccompanied.Comment: This has now been clarified in the discussion.
Reviewer 3 Report
This manuscript is not suitable for publishing on a peer-reviewed journal. It is too short and reads like a report rather than an article. There is no research gap nor innovative part addressed in the manuscript. It also lack spatial or environmental analysis.
Author Response
Dear reviewer.
We are well aware of the fact that this is a very humble study with obvious limitations. Despite these shortcomings we believe that this information is needed in the discussion about screening policies in Europe. Recently one of us (Hjern) has had the the task of reviewing the literature on health care for migrant children for WHO Europe. To his surprise there no recent material was found that provided a more comprehensive picture of the health care needs of the recently settled migrant children. Thus, we believe that this humble study can contribute to a more informed discussion about screening strategies for newly settled refugee children.
In this revised version of the article you will find that we have developed the methods and discussion sections considerably to better meet the standards of a peer review article.
Reviewer 4 Report
I really enjoyed reading the article. I believe it is well-written and well-structured and it provides interesting insights in particular for practitioners and policy-makers.
Minor remarks: not all the supplementary file is translated in English
Author Response
Thank you for your time and your nice comments!
Round 2
Reviewer 2 Report
Dear Authors
I appreciate the time taken to consider my comments. I spotted one spelling error in the Discussion (it says Ruropean instead of European). Otherwise, I enjoyed reading the revised version of the manuscript.
Reviewer 3 Report
Thanks for addressing the previous reviewer's comments in the extended discussion and method part. However, the innovation and filling of the research gap in this manuscript are not strong enough to be published as a peer-reviewed article.